# Effects of Solid-State Fermentation Pretreatment with Single or Dual Culture White Rot Fungi on White Tea Residue Nutrients and In Vitro Rumen Fermentation Parameters

Qi Yan [1,†], Miao Lin [1,2,3,†], Yinghao Huang [1], Osmond Datsomor [1], Kuopeng Wang [1] and Guoqi Zhao [1,2,3,*]

1   College of Animal Science and Technology, Yangzhou University, Yangzhou 225009, China
2   Institute of Agricultural Science and Technology Development, Yangzhou University, Yangzhou 225009, China
3   Joint International Research Laboratory of Agriculture and Agri-Product Safety, The Ministry of Education of China, Yangzhou University, Yangzhou 225009, China
*   Correspondence: gqzhao@yzu.edu.cn
†   These authors contributed equally to this work.

**Abstract:** Fermentation of agricultural by-products by white rot fungi is a research hotspot in the development of ruminant feed resources. The aim of this study was to investigate the potential of the nutritional value and rumen fermentation properties of white tea residue fermented at different times, using single and dual culture white rot fungal species. *Phanerochaete chrysosporium*, *Pleurotus ostreatus*, and *Phanerochaete chrysosporium* + *Pleurotus ostreatus* (dual culture) solid-state fermented white tea residue was used for 4 weeks, respectively. The crude protein content increased significantly in all treatment groups after 4 weeks. Total extractable tannin content was significantly decreased in all treatment groups ($p < 0.01$). *P. chrysosporium* and dual culture significantly reduced lignin content at 1 week. The content of $NH_3$-N increased in each treatment group ($p < 0.05$). *P. chrysosporium* treatment can reduce the ratio of acetic to propionic and improve digestibility. Solid state fermentation of white tea residue for 1 week using *P. chrysosporium* was the most desirable.

**Keywords:** white rot fungal; rumen fermentation; white tea residue; solid-state fermentation

## 1. Introduction

Protecting the environment is now a top priority for global development. The sustainable development of the biological recycling economy has become a hot research topic in the world [1]. In 2018, the United Nations Intergovernmental Panel on Climate Change (IPCC) set the goal of "carbon neutrality", limiting the rise in global temperature to 1.5 °C by the end of the 21st century [2]. For the past few years, by-products produced by agriculture and the food processing industry have threatened the environment, and these by-products are usually hard for the environment to degrade [3]. A growing world population necessitates a massive increase in food production. Global agricultural production is anticipated to be 60% higher in 2050 than in 2005, with the pressures on the environment and natural resource depletion expected to continue to escalate [4]. Therefore, it is very important to treat and utilize their by-products, which have positive benefits for environmental protection and economic benefit.

Tea originated in China, and tea drinking can be traced back to 4000 years ago. In The Qin and Han Dynasties, tea as a drink developed rapidly [5]. According to the Food and Agriculture Organization of the United Nations (FAO), 6.49 million tons of tea are produced globally yearly [6]. Tea production generates billions of tons of tea residue annually, mostly disposed of by direct burial and open-air incineration, polluting the environment [7]. Tea residue generally refers to pruned tea leaves, waste tea leaves, and tea stalks. The residues have similar nutrients and bioactive components, such as polyphenols, vitamins, minerals, terpenoids, pigments, amino acids, and polysaccharides, which are beneficial to animals

and can potentially be developed as animal feed [8]. Tea residue is mainly composed of lignin, cellulose, hemicellulose, protein, and polysaccharides [9]. Lignin, the largest non-carbohydrate in wood cellulose, provides strength and hydrophobicity to plant cell walls, and is difficult for microorganisms to degrade [10]. However, animals cannot absorb lignin, as it will alter the digestion and absorption of cellulose and hemicellulose [11].

Presently, there are many methods for lignin pretreatment to enhance the utilization value of by-products, including physical and chemical methods; biological pretreatment is considered the most effective method, as it is ecofriendly. The pretreatment method employing white rot fungi attracts the most attention, because they are the only organism capable of converting lignin into carbon dioxide and water.

*Pleurotus ostreatus* is attractive for the pretreatment of agricultural by-products and improving fibrous crop residues. It has been shown to secrete a large amount of lignin-degrading enzymes, which effectively degrade lignin while utilizing less cellulose [12]. *Phanerochaete chrysosporium* exhibits good lignin selective degradation characteristics and a faster colonization rate, which reduces the risk of fermentation substrate contamination and the fermentation pretreatment cycle. In a previous study by Luo et al. [13], the coculture of *P. chrysosporium* and *I. lacteus* had a 26.4% degradation rate of lignin, which was higher than the 23% of *P. chrysosporium* alone and the 21.2% of *I. lacteus* alone. Single fungal treatment of by-products is often time-consuming and has low biotransformation efficiency. However, fungal co-cultivation or dual culture may be an efficient pathway for lignin degradation. White rot fungi pretreatment of agricultural by-products as animal feed materials is an effective method for degrading lignin and upgrading the nutritional value of agricultural by-products [14]. However, there are no studies on white rot fungal pretreatment of tea residue for its potential utilization as animal feedstuff. It was hypothesized that using single and dual white rot fungi under solid-state fermentation of white tea residue could reduce lignin and improve digestibility.

In the present study, single strains *P. ostreatus* and *P. chrysosporium*, as well as its dual culture, were employed for the aerobic pretreatment of white tea residue via solid-state fermentation. The aim was to study the most effective strains and treatment times for the pretreatment of tea residue and the effect on rumen fermentation characteristics.

## 2. Materials and Methods

### 2.1. Fungal Source

*P. ostreatus* CGMCC 5.374 and *P. chrysosporium* CGMCC 5.829 were procured from China General Microbiological Culture Collection Center (CGMCC). The fungi were grown on a malt extract agar (MEA) plate culture medium (malt extract 20 g; ager 20 g; nutritional yeast 2 g; per L) and stored at 4 °C [15]. Agar plates were prepared using an autoclaved sterilized MEA (malt extract 20 g; ager 20 g; nutritional yeast 2 g; per L; 121 °C for 25 min), inoculated with a 0.5 cm$^2$ piece of the fungi and incubated at 25 ± 1 °C until mycelium covered the entire surface of the plates.

### 2.2. Millet Spawn Preparation

Millet grains were washed in water and boiled for 15 min. The boiled grains were transferred onto a sieve to drain. The grain was packed into two polyethylene bags (30 × 40 cm) until it was three-quarters full, and then autoclaved at 121 °C for 20 min. The content of each bag was permitted to cool to room temperature, and then separately inoculated aseptically with five 1 cm$^2$ of mycelium ager and sealed. The contents of the polyethylene bags were shaken manually to ensure uniform mixing of the mycelium with the grains. It was then incubated in a ventilated incubator (Jing Hong Co., Ltd., Changzhou, China) at 25 ± 1 °C and 60% humidity until the mycelia colonized all the grains. The spawns were then removed, allowed to cool, and stored in a cold room at 4 °C to stop the mycelia from further growth and to keep for future use.

### 2.3. Tea Residue Source

White tea residue was provided by Shenshan tea cooperation, located in Zhenghe County, Fujian, China. The tea residue was physically cut into 2 to 3 cm lengths. Polypropylene bags were filled with 200 g of white tea residue, and the moisture content was adjusted to 70% using distilled water. The polypropylene bags were then autoclaved, sterilized at 121 °C for 20 min, and allowed to cool at room temperature.

### 2.4. Fermentation Method

The bags were inoculated with millet spawn at 6% (*w/w*) of straw, *P. ostreatus* 6%, *P. chrysosporium* 6%, and dual culture *P. ostreatus* 3% + *P. chrysosporium* 3%. The control was prepared in the same way as the treatment groups, except for the spawn inoculum. The inoculated bags were shaken to ensure uniform spawn distribution, then incubated at a temperature of 25 ± 1 °C and 75–80% humidity. The bags were incubated at 24 °C and a relative humidity of 70%. Each group was sampled at 1, 2, 3, and 4 weeks. The weekly samples consisted of treated substrate and mycelium, which were oven-dried at 65 °C for 48 h. The dried fungi-treated white tea residue was ground over a 1 mm sieve using a miller machine (CM100, Yongguangming Co., Ltd., Beijing, China) to obtain a homogenous sample, then was stored for further chemical and in vitro analysis.

### 2.5. Chemical Index and Fiber Carbohydrates Analysis

Weekly samples were dried in an oven drier (DHG-9123A, Wollen Instrument Equipment Co., Ltd., Shanghai, China) at 105 °C for 3 h to determine their dry matter content (DM) [16]. The nitrogen (N) content and ether extract (EE) were determined using the Kjeldahl and Soxhlet method, according to the Association of Official Analytical Chemists [17]. The crude protein (CP) was estimated by multiplying the N by 6.25. Ash content was measured according to Jiang et al. [18], carbonized in a muffle furnace at 550 °C for 3 h. Neutral detergent fiber (NDF), acid detergent fiber (ADF), and acid detergent lignin (ADL) were measured using the methods described by Van Soest et al. [16], and an optical fiber analyzer (2000i, ANKOM Technology, Macedon, NY, USA). Hemicellulose (HC) content is the difference of NDF minus ADF. Cellulose (CL) content was calculated as the difference between ADF and ADL. The total extractable tannins were determined according to the method of Royer M et al. [19].

### 2.6. Morphological Observation

A scanning electron microscope was used to analyze the morphological structure of tea residue treated with fungi. The tea residue was first subjected to gold spray treatment using an ion sprayer (SCD500, Baltec CO., Ltd., Canton of Zurich, Switzerland) and then the structure of the tea residue was observed using a scanning electron microscope (SEM) (S-4800, Hitachi Co., Ltd., Tokyo, Japan). The role of butyrate as part of a milk replacer and starter diet was studied.

### 2.7. In Vitro Fermentation

Holstein cows were selected as the experimental animals. A total of 3 cows were selected, all of which were equipped with rumen fistula. The weight of the experimental cows was similar (650 ± 25 kg); they had been pregnant twice and were in good health. Cows were fed in equal amounts three times a day, with tethered feeding and free drinking water; the enclosure was disinfected regularly with disinfectants such as Bromogeramine. The feces was cleaned and routinely discharged. The cow house was kept clean and dry throughout the test. The maintenance of the Holstein cows' fistulas and rumen fluid collection procedures was approved by the Animal Care Committee of Yangzhou University (Jiangsu, China).

According to the method of Menke et al. [20] and Lin et al. [21], fresh rumen liquid was obtained from the rumen fistula of cows, filtered into the heat preservation device through a sterilized cotton cloth, mixed with artificial buffer 1:2 (*v/v*), mixed evenly with a

magnetic mixer at $8000 \times g$, and washed continuously with $CO_2$. A total of 220 mg of the air-dried sample was placed into a fermentation bottle with a volume of 128 mL, and 30 mL of the above artificial rumen liquid was injected. The mouth of the fermentation bottle is tightly plugged with a rubber stopper and fixed with an aluminum cap to ensure complete sealing. The fermentation bottles were cultured at 39 °C in an incubator (THZ-320, Jinghong Devices, Shanghai, China) with a vibration frequency of 150 rpm for 72 h. The blank group (only including artificial rumen liquid), the control group, and the experimental group were repeated by four.

The air pressure in each fermentation bottle was measured with an air pressure detector (DPG1000B15PSIG-5, Cecomp Electronics, Libertyville, IL, USA) at 0, 2, 4, 8, 12, 24, 36, 48, and 72 h. The gas volume was calculated using the formula $Vgas = Vj \times Ppsi \times 0.068004084$, where Vgas is the gas volume at 39 °C, mL; Vj is the volume of the fermentation bottle with the artificial rumen liquid removed, mL; Psi is the pressure of the vial, psi; and 0.068004084 is the conversion factor.

After incubation for 72 h, the fermentation bottle was taken out of the incubator and placed in an ice-water bath to terminate the fermentation. Then the pH of the fermentation mixture was measured with a pH tester (PHS-25, instrument and electricity Co., Ltd., Shanghai, China). The fermentation mixture was transferred to a centrifuge tube and then centrifuged at $8000 \times g$ and 4 °C for 10 min. The fermentation mixture was transferred to a centrifuge tube, and then centrifuged at $8000 \times g$ and 4 °C for 10 min, which was the solid–liquid separation of the fermentation mixture. In a 2 mL centrifuge tube, 1 mL of supernatant was combined with 0.2 mL of 20% metaphosphoric acid, then the orifice of the centrifuge tube was sealed with a sealing film and stored overnight at 4 °C for subsequent determination of volatile fatty acid (VFA) concentration. Using an Agilent capillary gas chromatographic column with a thermal conductivity detector (30 m × 0.32 mm × 0.25 μm), the concentration of VFA was determined by gas chromatography–mass spectrometry (GC-MS 9800, Shanghai Kechuang Chromatographic Instrument Co., Ltd., Shanghai, China). The injector, chromatographic column, and detector temperatures were 200 °C, 110 °C and 200 °C, respectively. The carrier gas is nitrogen, the flow rate is 50 mL/min, and the injection volume is 1 μL. $NH_3$-N is determined by the phenol–sodium hypochlorite method. First, 50 μL of the above-centrifuged supernatant were placed in a 10 mL test tube. Then, 2.5 mL of phenol reagent and 2.0 mL of sodium hypochlorite reagent were added to form a mixed solution. The mixture was reacted in a water bath at 95 °C for 5 min. After cooling, a spectrophotometer was used for colorimetric determination at the wavelength of 630 nm.

*2.8. Statistical Analysis*

Data were initially collated using Excel 2016 (Microsoft Corp., Redmond, WA, USA) and then analyzed using SPSS version 26.0 (IBM Corp., Armonk, NY, USA). Proximate composition and cell wall composition data were subjected to two-way ANOVA, including fixed effects of the fungal treatment group (T), treatment time (D), and treatment time interaction (T × D). Gas production, pH, $NH_3$-N, and VFA data were analyzed by one-way ANOVA. Post hoc, multiple comparisons were performed with the Duncan significance test at a significance level of 0.05 to determine significance between experimental groups.

### 3. Results

*3.1. Changes in Chemical Index and Fiber Carbohydrates*

The changes in proximate composition of the three treatment groups at different time points are shown in Table 1. Fungal treatment had a significant effect on the proximate composition of tea residue, and all treatment groups caused DM loss of tea residue ($p < 0.01$). At 4 weeks, the DM content of the dual culture treatment group was significantly higher than that of *P. ostreatus* and *P. chrysosporium* ($p < 0.05$). Regarding CP content, the CP of the dual culture group was significantly higher than that of the *P. ostreatus* treatment group at 1 week ($p < 0.05$). Furthermore, there was no significant difference among the three treatment groups at other time points, and the content of CP in the three treatment groups

was significantly higher at 4 weeks than at 0 weeks ($p < 0.01$). Although not always linear, the EE content of the three treatment groups was significantly higher at 4 weeks than at 0 weeks ($p < 0.01$), with the *P. chrysosporium* treatment group having a significantly higher EE content than *P. ostreatus* and the dual culture treatment group ($p < 0.05$).

**Table 1.** Proximate composition (%/DM) of white tea residue after 0, 1, 2, 3, and 4 weeks of fermentation with different fungi cultures.

| Items | Treatment | Duration of Treatment (Weeks) | | | | | SD | *p*-Value | | |
|---|---|---|---|---|---|---|---|---|---|---|
| | | 0 | 1 | 2 | 3 | 4 | | T | W | T × W |
| DM | *P. chrysosporium* | 96.62 [a] | 91.76 [b] | 92.94 [Bb] | 92.99 [Bb] | 92.54 [Bb] | 1.816 | <0.01 | <0.00 | 0.14 |
| | *P. ostreatus* | 96.62 [a] | 91.86 [c] | 93.84 [ABb] | 93.43 [ABb] | 93.48 [ABb] | | | | |
| | Dual | 96.62 [a] | 94.07 [b] | 93.89 [Ab] | 94.01 [Ab] | 94.33 [Ab] | | | | |
| CP | *P. chrysosporium* | 14.51 [c] | 14.61 [ABc] | 16.19 [b] | 16.18 [b] | 16.70 [a] | 1.154 | 0.24 | <0.01 | 0.10 |
| | *P. ostreatus* | 14.51 [c] | 14.30 [Bc] | 16.03 [b] | 16.70 [a] | 16.80 [a] | | | | |
| | Dual | 14.51 [d] | 15.30 [Acd] | 15.94 [bc] | 16.37 [ab] | 17.16 [a] | | | | |
| EE | *P. chrysosporium* | 3.85 [c] | 6.52 [b] | 6.56 [b] | 6.70 [b] | 8.22 [Aa] | 2.044 | <0.01 | <0.01 | 0.25 |
| | *P. ostreatus* | 3.85 | 5.55 | 5.76 | 5.70 | 5.25 [B] | | | | |
| | Dual | 3.85 [b] | 4.38 [ab] | 4.75 [ab] | 4.80 [ab] | 5.54 [Ba] | | | | |
| ASH | *P. chrysosporium* | 4.37 [b] | 4.43 [b] | 5.56 [ABa] | 5.69 [a] | 5.90 [a] | 0.828 | <0.05 | <0.01 | 0.23 |
| | *P. ostreatus* | 4.37 [e] | 4.72 [d] | 5.30 [Bc] | 5.72 [b] | 6.30 [a] | | | | |
| | Dual | 4.37 [d] | 5.48 [c] | 5.76 [Abc] | 6.07 [ab] | 6.29 [a] | | | | |
| TET | *P. chrysosporium* | 5.73 [a] | 2.39 [Bb] | 1.53 [c] | 1.22 [cd] | 0.81 [d] | 1.822 | <0.01 | <0.01 | <0.01 |
| | *P. ostreatus* | 5.73 [a] | 3.88 [Ab] | 2.95 [bc] | 2.35 [c] | 0.90 [d] | | | | |
| | Dual | 5.73 [a] | 3.57 [Ab] | 1.60 [c] | 1.42 [c] | 1.03 [c] | | | | |

Note: 0 weeks, only sterilized control. SD, standard deviation of the mean; [a–e] means in the same row with different superscripts differ ($p < 0.05$); [A,B] means in the same column with different superscripts differ ($p < 0.05$); T, the effect of fungi treatment species; W, the effect of treatment week; T × W, the interaction between fungi treatment and treatment week. DM, dry matter; CP, crude protein; EE, ether extract; TET, total extractable tannins.

At 4 weeks, the EE content of the three treatment groups was significantly higher than at 0 weeks ($p < 0.01$), and the EE content of the *P. chrysosporium* treatment group was significantly higher than that of *P. ostreatus* and the dual culture group ($p < 0.05$). In terms of ash content, the *P. chrysosporium* treatment group and the dual culture treatment group were significantly higher than the *P. ostreatus* treatment group at 2 weeks ($p < 0.05$). However, with the extension of time, there was no significant difference between the fourth week and the three groups, which were significantly higher than that of the 0 week ($p < 0.01$). The TET of all treatment groups decreased linearly with time ($p < 0.01$). However, the *P. chrysosporium* treatment group had the fastest decline at 1 week.

In this study, the NDF content of the *P. chrysosporium* treatment group decreased at 1 week ($p < 0.01$) (Table 2). However, with the extension of time, the content of NDF increased gradually ($p < 0.01$), reaching 55.78% in the 4 weeks. The NDF of the *P. ostreatus* treatment group increased to 36.76% at 1 week, but was not significant. The *P. ostreatus* NDF content of 40.23%, 41.45%, and 41.76% at 2, 3, and 4 weeks were significantly higher than 35.84% at 0 weeks ($p < 0.01$). The same trend appeared in the dual culture and *P. ostreatus* treatment groups. There was no significant difference between 0 weeks and 1 week, but the contents of 42.15%, 48.77%, and 53.11% in the 2, 3, and 4 weeks were significantly higher than 35.84% at 1 week ($p < 0.01$). At 4 weeks, the NDF content of the three treatment groups was *P. chrysosporium* > dual culture > *P. ostreatus* ($p < 0.01$).

**Table 2.** Cell wall composition (%/DM) of white tea residue after 0, 1, 2, 3, and 4 weeks of fermentation with different fungi cultures.

| Items | Treatment | Duration of Treatment (Weeks) | | | | | SD | *p*-Value | | |
|---|---|---|---|---|---|---|---|---|---|---|
| | | **0** | **1** | **2** | **3** | **4** | | **T** | **W** | **T × W** |
| NDF | *P. chrysosporium* | 35.84 ᵉ | 33.22 ᴮᵈ | 39.64 ᴮᶜ | 51.39 ᴬᵇ | 55.78 ᴬᵃ | 7.04 | <0.01 | <0.01 | <0.01 |
| | *P. ostreatus* | 35.84 ᵇ | 36.76 ᴬᵇ | 40.23 ᴮᵃ | 41.45 ᶜᵃ | 41.76 ᶜᵃ | | | | |
| | Dual | 35.84 ᵈ | 36.34 ᴬᵈ | 42.15 ᴬᶜ | 48.77 ᴮᵇ | 53.11 ᴮᵃ | | | | |
| ADF | *P. chrysosporium* | 29.75 ᵇ | 26.10 ᴮᶜ | 30.16 ᴮᵇ | 39.96 ᴬᵃ | 42.07 ᴬᵃ | 5.27 | <0.01 | <0.01 | <0.01 |
| | *P. ostreatus* | 29.75 ᶜ | 27.95 ᴬᵈ | 30.53 ᴮᵇᶜ | 32.05 ᶜᵇ | 34.07 ᴮᵃ | | | | |
| | Dual | 29.75 ᵈ | 28.01 ᴬᵈ | 32.26 ᴬᶜ | 36.59 ᴮᵇ | 40.87 ᴬᵃ | | | | |
| ADL | *P. chrysosporium* | 9.05 ᶜ | 6.73 ᶜᵉ | 8.27 ᴮᵈ | 13.94 ᴬᵇ | 14.71 ᴬᵃ | 2.40 | 0.61 | <0.01 | <0.01 |
| | *P. ostreatus* | 9.05 ᶜ | 9.19 ᴬᶜ | 10.77 ᴬᵇ | 10.90 ᴮᵇ | 12.06 ᶜᵃ | | | | |
| | Dual | 9.05 ᶜ | 7.56 ᴮᵈ | 10.59 ᴬᵇ | 11.54 ᴮᵇ | 13.15 ᴮᵃ | | | | |
| HC | *P. chrysosporium* | 7.29 ᶜ | 7.11 ᴮᶜ | 9.48 ᵇᶜ | 11.43 ᵃᵇ | 13.70 ᴬᵃ | 2.65 | <0.01 | <0.05 | <0.01 |
| | *P. ostreatus* | 7.29 ᵇ | 8.81 ᴬᵃᵇ | 9.70 ᵃ | 9.39 ᴮᵃ | 7.69 ᴮᵇ | | | | |
| | Dual | 7.29 ᵈ | 8.32 ᴬᶜ | 9.89 ᵇ | 12.18 ᴬᵃ | 12.24 ᴬᵃ | | | | |
| CL | *P. chrysosporium* | 20.70 ᵇ | 19.36 ᴬᴮᵇ | 21.89 ᴬᵇ | 26.01 ᴬᵃ | 27.36 ᴬᵃ | 3.50 | <0.01 | <0.01 | <0.05 |
| | *P. ostreatus* | 20.70 ᵃᵇ | 18.75 ᴮᶜ | 19.76 ᴮᵇᶜ | 21.14 ᶜᵃᵇ | 22.00 ᴮᵃ | | | | |
| | Dual | 20.70 ᶜ | 20.42 ᴬᶜ | 21.66 ᴬᶜ | 25.04 ᴮᵇ | 27.71 ᴬᵃ | | | | |

Note: 0 weeks, only sterilized control. SD, standard deviation of the mean; ᵃ⁻ᵉ means in the same row with different superscripts differ ($p < 0.05$); ᴬ⁻ᶜ means in the same column with different superscripts differ ($p < 0.05$); T, the effect of fungi treatment species; W, the effect of treatment week; T × W, the interaction between fungi treatment and treatment week. NDF, neutral detergent fiber; ADF, acid detergent fiber; ADL, acid detergent lignin; HC, hemicellulose; CL, cellulose.

## 3.2. Morphological Observation

Figure 1 illustrates the structural differences of the tea residue after each fungi treatment. The *P. chrysosporium* treatment group had the greatest degree of structural damage, the largest pores, and many attachments on the fiber surface in Figure 1B. The *P. ostreatus* treatment group and the dual treatment group showed smaller pores. However, the fiber surface was still consistent with the *P. chrysosporium* treatment group, with more attachments, in Figure 1C,D.

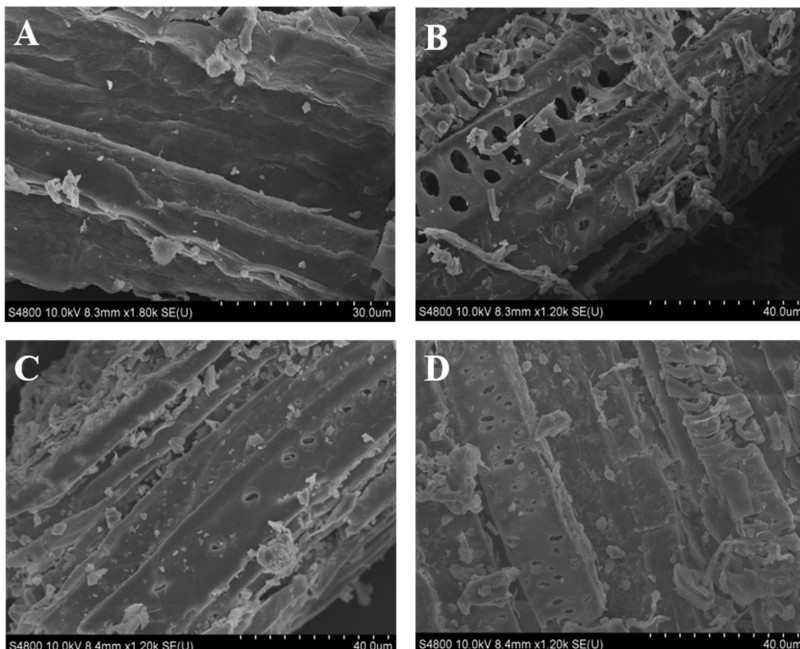

**Figure 1.** Morphological observation of tea residue. (**A**) Showing surface image of only sterilized control; (**B**) P. chrysosporium treatment group; (**C**) P. ostreatus treatment group; (**D**) dual treatment group.

### 3.3. In Vitro Fermentation

There were significant differences between different treatment groups at 2 h of gas production ($p < 0.05$) (Table 3). After 72 h of fermentation, the gas production of the control group, the *P. chrysosporium* treatment group, the *P. ostreatus* treatment group, and the dual culture treatment group was 131.1 mL/220 mg DM, 104.32 mL/220 mg DM, 113.38 mL/220 mg DM, and 118.17 mL/220 mg DM, respectively. The gas production of the three fungal treatment groups was significantly lower than that of the control group ($p < 0.05$).

**Table 3.** In vitro gas production from white tea residue after incubation with different fungi cultures.

| Items | Control | *P. chrysosporium* | *P. ostreatus* | Dual | SD | *p*-Value |
|---|---|---|---|---|---|---|
| GP 2 h | 46.57 | 53.46 | 49.6 | 50.43 | 7.11 | 0.382 |
| GP 4 h | 76.53 [a] | 71.05 [ab] | 62.27 [c] | 68 [bc] | 7.51 | <0.05 |
| GP 8 h | 97.63 [a] | 82.51 [b] | 75.92 [b] | 82.58 [b] | 10.41 | <0.05 |
| GP 12 h | 106.26 [a] | 91.49 [b] | 88.26 [b] | 98.07 [ab] | 10.44 | <0.05 |
| GP 24 h | 116.62 [a] | 96.04 [b] | 102.32 [b] | 107.48 [ab] | 10.45 | <0.05 |
| GP 36 h | 119.15 [a] | 97.7 [b] | 106.77 [ab] | 113.53 [a] | 13.71 | <0.05 |
| GP 48 h | 122.88 [a] | 100.04 [b] | 109.84 [ab] | 115.92 [a] | 15.19 | <0.05 |
| GP 72 h | 131.1 [a] | 104.32 [b] | 113.38 [b] | 118.17 [ab] | 15.29 | <0.05 |

Note: [a–c] means in the same row with different superscripts differ ($p < 0.05$); GP, gas production; h represents the time of gas production in hours.

As shown in Table 4, the pH values of each group in this experiment were within the normal rumen pH range. The pH values of the *P. chrysosporium* treatment group and the dual culture treatment group were lower than those of the control group ($p < 0.05$). The order of NH$_3$-N content from high to low was *P. ostreatus* treatment group > dual culture treatment group > *P. chrysosporium* treatment group > control group, whereby the three fungal treatment groups were significantly higher than the control group ($p < 0.05$). The total VFA and acetate acid in the three fungal treatment groups were significantly lower than in the control group ($p < 0.05$). Butyric acid, isobutyric acid, butyric acid, and valeric acid among different treatment groups were significantly different ($p < 0.01$). The propionate acid of the *P. chrysosporium* treatment group and the control group was significantly higher than that of the *P. ostreatus* treatment group and the dual treatment group ($p < 0.01$). The isobutyric acid of the *P. chrysosporium* treatment group and the control group was lower than that in the other two fungal treatment groups ($p < 0.01$). Butyrate acid of the *P. chrysosporium* treatment group was the lowest. The A/P of the *P. chrysosporium* treatment group was the lowest among all the treatment groups. *P. ostreatus* and dual treatment groups were higher than the control group ($p < 0.01$).

**Table 4.** Fermentation products of the control and the white tea residue with different treatments after 72 h of digestion by rumen microbiota in vitro.

| Items | Control | *P. chrysosporium* | *P. ostreatus* | Dual | SD | *p*-Value |
|---|---|---|---|---|---|---|
| pH | 6.35 [a] | 6.28 [b] | 6.31 [ab] | 6.29 [b] | 0.04 | <0.05 |
| NH$_3$-N, mg/dL | 12.61 [b] | 18.06 [a] | 20.59 [a] | 19.74 [a] | 4.79 | <0.05 |
| TVFA, mmol/L | 114.02 [a] | 107.31 [b] | 103.16 [b] | 104.68 [b] | 7.36 | <0.05 |
| Individual VFA (mmol/L VFA) | | | | | | |
| Acetic acid | 74.48 [a] | 69.10 [b] | 66.10 [b] | 66.99 [b] | 5.77 | <0.05 |
| Propionic acid | 20.27 [a] | 20.25 [a] | 16.5 [b] | 16.85 [b] | 1.43 | <0.01 |
| A/P | 3.67 [b] | 3.41 [c] | 4.00 [a] | 3.97 [a] | 0.28 | <0.01 |
| Isobutyric acid | 0.88 [b] | 0.76 [b] | 1.40 [a] | 1.36 [a] | 0.51 | <0.01 |
| Butyric acid | 12.63 [b] | 12.07 [c] | 14.33 [a] | 14.79 [a] | 0.98 | <0.01 |
| Isovaleric acid | 3.57 [a] | 3.20 [ab] | 3.16 [ab] | 3.01 [b] | 0.25 | 0.10 |
| Valeric acid | 2.16 [a] | 1.92 [b] | 1.64 [c] | 1.66 [c] | 0.29 | <0.01 |

Note: [a–c] means in the same row with different superscripts differ ($p < 0.05$); VFA, volatile fatty acid; TVFA, total volatile fatty acid; A/P, acetic to propionic acid ratio.

## 4. Discussion

In the current study, the DM content of the three fungi treatment groups decreased linearly with time, which was consistent with previous studies [22]. In the study conducted by Yang et al. [23], the CP content of all camellia seed residues treated with fungi increased significantly. In the current study, the CP content of the three treatment groups at 2, 3, and 4 weeks was significantly higher than that of the untreated group (control). This may be because fungi can synthesize proteins from nitrates and amine in fermentation substrates [24]. Another potential factor is the increase in chitin, a component of fungal cell walls during fermentation, composed of N-acetyl and β-glucan, which also contains N [25]. The growth of fungi will lead to the increase in chitin content. The EE content of the three fungal treatment groups was significantly increased after treatment, which was consistent with previous studies. The increase in EE of fungi-treated residue originates from fungi biomass. Fungi derive nutrients by decomposing and transforming OM into substances, including lipids [26]. Lipid compounds play vital functions in the fungi life cycle, occurring predominantly in the stationary growth phase [27]. The ash content of tea residue increased significantly, which is similar to the research of Nayan, N et al. [28]. For this result, we speculate that the first is related to the consumption of organic matter (OM) by fungi during the fermentation process, and the second is related to the increase in ash content due to the release of minerals and inorganic substances in the residual matrix by the fungi [29]. Versatile peroxidase (VP) is a lignin-degrading enzyme produced by white rot fungi with broad substrate preferences, which can degrade phenolic and non-phenolic compounds [30]. The decrease in total extractable tannin content in the three treatment groups can be attributed to VP, since tannin is a phenolic compound [31].

Previous studies have shown that NDF, ADF, and ADL contents in the substrates generally decreased with time during the fungal fermentation treatment [29]. However, in the current study, NDF, ADF, and ADL contents increased in all treatment groups over time, although not in a linear trend. We hold the opinion that the fungal fermentation substrate combination is responsible for this phenomenon. Firstly, white rot fungi's growth and degradation selectivity differed among different species and batches of substrates. Labusehagne et al. [32] reported that the growth of *P. ostreatus* on 15 different batches of wheat straw varied widely, with yields ranging from 123 kg to 262 kg per ton of substrate. Secondly, differences between strains can also be observed across species. Membrillo et al. [33] reported that two different strains of *P. ostreatus* produced a maximum four-fold difference in laccase and a maximum three-fold difference in xylanase, using bagasse as substrate. It has been reported that the selectivity of white rot fungi for lignin degradation may be related to the nitrogen content in the substrate. The lignin degradation selectivity of white rot fungi is usually more easily stimulated in substrates with low nitrogen sources. In other words, relatively high nitrogen content can inhibit their lignin degradation selectivity while stimulating the growth and consumption of soluble carbohydrates [34,35]. The CP content of the wheat straw used in the previous study by Niu et al. [36] was 4.42% lower than that of the white tea residue in this study, which was 14.51%. In their study, solid-state fermentation of *P. chrysosporium* and *P. ostreatus* was observed. The contents of NDF, ADF, and ADL in the wheat straw decreased with the time of fungal fermentation treatment. This suggests that this study's relatively high CP content in white tea residue may induce the lower selectivity of fungal lignin degradation. When fungi grow in large quantities, they secrete a large number of hydrolytic enzymes which help them consume fermentable carbohydrates as a carbon source, in order to meet their growth and metabolic needs. This view is supported by the increasing fibrous carbohydrate content at 2 weeks in this study's three fungal treatment groups, which is the main reason for the increased ADL content in the three groups. White rot fungi mainly have two sets of enzymatic hydrolysis systems, including the lignin oxidase system responsible for degrading lignin and the hydrolase system responsible for degrading polysaccharides. There are three main enzymes in the lignin oxidase system, namely Lignin peroxidase (Lip), Versatile peroxidase (VP), manganese peroxidase (MnP) and Laccase (Lac) [37,38]. *P. ostreatus* can produce the above three

lignin-degrading enzymes; it has high fibrous carbohydrate hydrolysis capacity and lignin hydrolysis activity, and has been widely studied in the biotreatment of crop residue into animal feed. However, because it simultaneously produces lignin-degrading enzymes, cellulose, and hemicellulose-degrading enzymes during the fermentation process, it is termed as a moderately selective fungi, and thus, not highly selective for the degradation of lignin [39,40]. In the present study, *P. ostreatus* did not selectively degrade lignin in the substrate at any stage of fermentation, but instead continued to degrade soluble carbohydrates, stimulating white rot fungal growth and consumption of soluble carbohydrates [34,35]. This may be related to the fact that the aforementioned higher nitrogen content does not stimulate the lignin selectivity of fungi. Not all white rot fungi produce the three lignin-degrading enzymes mentioned above. *P. chrysosporium* is considered to be a highly efficient lignin-degrading strain, because it produces a large amount of lignin-degrading enzymes, such as Lip, VP, and MnP, during fermentation, but *P. chrysosporium* usually does not produce Lac [37]. Moreover, *P. chrysosporium* has a rapid colonization rate and can degrade lignin rapidly over a few days [31]. The present study also observed similar growth characteristics, with the *P. chrysosporium* treatment group degrading the ADL of tea residue in the early fermentation stage, resulting in a small loss of cellulose and hemicellulose. This is due to the fact that the enzymes produced by *P. chrysosporium* include various glucoside hydrolases (GHs), which are involved in the degradation of cellulose and hemicellulose [41,42]. In our current research, the change in the trend of cell wall compositions in the dual treatment group during solid-state fermentation was similar to that of the *P. chrysosporium* treatment group. We speculate that this is due to the change in the cell wall composition of the tea residue by the dual culture treatment group, whereby *P. chrysosporium* played a major role. Van kuijk et al. [43] previously stated that, in the process of dual-cultivation of fungi, the contribution of each fungus to the co-culture is unclear, so only one fungus may be left to play a role in the co-culture. This is consistent with our point of view.

The plant cell wall is composed of a multi-layered structure consisting of three types of layers, namely the intermediate layer (M), the primary layer (P), and the secondary layer (S). Among these, the S layer is usually composed of three sub-layers, including the outer layer (S1), the middle layer (S2), and the inner layer (S3) [38]. Cellulose, hemicellulose and lignin have different distributions in these layers, and the corresponding contents are also different. For example, in plant fibers, the cellulose content gradually increases from the M layer to the S layer, and the cellulose content of the S2 layer and the S3 layer is the highest [44]. *P. chrysosporium* produced larger pores than *P. ostreatus* and the dual culture. There are two possible reasons which explain this result. *P. chrysosporium* is a typical non-selective fungus. It decomposes cellulose, hemicellulose, and lignin simultaneously. Secondly, it is known that *P. chrysosporium* grows very quickly, so it consumes more cell wall components for its own growth metabolism. During the entire fermentation process, *P. ostreatus* degraded cellulose at 1 week, so we believe that the pores observed may result from cellulose removal by *P. ostreatus*. The pores produced by the dual culture are smaller. We believe that a kind of competition causes it, and the two fungi will be more inclined to take in soluble carbohydrates as a carbon source for growth and metabolism in the competition. According to the results of the degradation of cell wall components, the dual culture only effectively degraded ADL at 1 week, but did not degrade cellulose or hemicellulose. Scanning electron microscopy (SEM) is an important microscopic imaging technique that can directly observe morphological features [45]. Through SEM, we cannot accurately determine which part of the initially removed hole is part of cellulose, lignin, or hemicellulose. It may be two or even all of them removed at the same time. Through SEM, the overall damage caused by the fungi to the cell wall can be seen to judge the effect of various fungi enzymes, such as lignin-degrading enzymes and cellulose hydrolases on the cell wall. The tea residue with white rot fungi produced large changes, such as a loose surface, curved cracks, enlarged specific surface area, and even cracking into small cross-sections. This is strong evidence that the tea residue had an altered cell wall structure.

In vitro gas production is widely used in feed evaluation experiments [46]. In the current study, in vitro gas production technology was used to evaluate the gas production changes of three white rot fungi treatments with tea residue for 72 h. During the early stages of the in vitro gas production process, the soluble fraction in the feed usually contributes largely to gas production. In this study, the *P. chrysosporium* treatment group had higher gas production at 4 h than the other two fungal treatment groups, which was consistent with the report by Niu et al. [36]. With subsequent time intervals, the gas production of the three fungal treatment groups was lower than that of the control group, especially the *P. chrysosporium* treatment group, which may be related to the metabolite Lovastatin during fungal growth. Lovastatin is a competitive inhibiter of HMG-CoA reductase, which is a key enzyme in the cell membrane synthesis of methanogenic archaea, thereby inhibiting the activity of methanogenic archaea and reducing $CH_4$ production [47,48]. A study of gas composition was not carried out in this study, but will be a potential research work in the future.

The pH value is an important indicator reflecting the rumen function of ruminants, such as the types of rumen fermentation products and the influence of many factors, such as the precipitation of organic acids [49]. In this study, the *P. chrysosporium* treatment group and the dual culture treatment group were significantly lower than the control group, but both were within the appropriate range and would not affect the normal function of the rumen [50]. In addition, low pH is usually associated with a high feed conversion rate [51]. In this study, the *P. chrysosporium* treatment group and the dual culture treatment group reduced the ADL content of tea residue at 1 week. Therefore, we speculate that the fungi altered the cell wall structure, which dissociates tight junctions between cellulose, hemicellulose, and lignin, making it easier for rumen microbes to access and utilize rumen fermentable carbohydrates, such as cellulose and hemicellulose. The normal range of $NH_3$-N content in the rumen is 6.3–27.5 mg/dL, and all groups in this study were within the standard range [52]. The $NH_3$-N in the rumen of ruminants is determined by the rate of production, its utilization by microorganisms, and its absorption by the entire rumen wall [53]. In the study conducted by Nishida et al. [54] and Kondo et al. [55], it was confirmed that the high tannin content in tea residue inhibits the rate of protein degradation in ruminants, thereby reducing $NH_3$-N. The $NH_3$-N content in this study's three fungal treatment groups was higher than that in the control group, which was caused by the reduction of TET content. The $NH_3$-N reflects the conversion of dietary nitrogen to microbial nitrogen, and is generally positively correlated with dietary protein utilization [56]. The increased $NH_3$-N in this study confirms the higher rate of ruminal protein degradation in tea residue treated with fungi.

The volatile fatty acids (VFAs) produced by rumen fermentation are the main energy source for the maintenance and growth of rumen microorganisms, mainly from the degradation of dietary carbohydrates [57,58]. It is evident that the total VFA content of the three fungal treatment groups was lower than that of the control group due to the loss of different types of carbohydrates in the three treatment groups over 1 week. Acetic acid, Propionic acid and butyric acid account for more than 95% of the total VFA concentration [59]. The acetic acid content of the *P. chrysosporium* treatment group decreased, and the propionic acid content was not significantly different from the control group. This may be due to the reduction of cell wall components. The acetic acid and propionic acid of the *P. ostreatus* treatment group and the dual culture treatment group were lower than those in the control group, which may be due to the fact that the TET content of the *P. ostreatus* treatment group and the dual culture treatment group was still higher during this fermentation period. This may have restricted the ability of some rumen microorganisms to utilize carbohydrates. Acetic acid is mostly dissipated in the form of heat during rumen digestion; propionic acid is an important precursor to ruminant gluconeogenesis. Generally, the lower the A/P ratio, the higher the digestibility of the feed [60,61]. In this experiment, the A/P ratio of the *P. chrysosporium* treatment group was the lowest, and it could be inferred that the tea residue treated with *P. chrysosporium* had the best digestibility. $NH_3$-N is a major metabolite

when rumen bacteria ferment amino acids, but other metabolites are also produced. When branched-chain amino acids (BCAA) such as Val, Leu, and Ile are oxidatively deaminated, they are converted to branched-chain volatile fatty acids (BCVFA) such as isobutyric acid, isovaleric acid, and 2-methyl butyrate, respectively. Although BCVFA is partly derived from the microbial protein cycle, it is mainly derived from the degradation of true proteins. Isobutyrate acid was higher in the *P. ostreatus* treatment group and the dual culture treatment group, which may be due to the rumen fermentation mode change. Some amino acids were converted to BCVFA.

Further in vivo digestion trials should be performed to determine the feasibility of utilizing fungi-treated tea residue as ruminant feed, as well as toxicological studies to determine its safety.

## 5. Conclusions

The use of *P. chrysosporium* in short-term solid-state fermentation of tea residue can effectively reduce the ADL content. It can also change the A/P ratio of tea residue in the rumen, reduce the TET content, and improve the protein degradation rate.

In conclusion, solid-state fermentation with *P. chrysosporium* for 1 week is the best treatment option. However, the problem of the sterilization step needs to be solved in order to facilitate the fast processing of tea residue or other problems for practical application as feed. In vivo digestion trials are also needed to determine safety as well as economic benefits.

**Author Contributions:** Conceptualization, Q.Y., M.L., K.W. and O.D.; data curation, Q.Y.; formal analysis, Q.Y.; methodology, Q.Y., Y.H., K.W. and O.D.; supervision, G.Z. and M.L.; validation, Q.Y., O.D., K.W. and M.L.; writing—original draft, Q.Y.; writing—review and editing, Q.Y. and O.D. All authors have read and agreed to the published version of the manuscript.

**Funding:** This study was supported by the earmarked fund for CARS36.

**Institutional Review Board Statement:** The animal study protocol was approved by the Institutional Review Board (Animal Care and Use Committee) of Yangzhou University (Jiangsu, China).

**Informed Consent Statement:** Not applicable.

**Data Availability Statement:** The data presented in this study are available on reasonable request from the corresponding authors.

**Conflicts of Interest:** The authors declare no conflict of interest.

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
