# Peer review of "Effects of Solid-State Fermentation Pretreatment with Single or Dual Culture White Rot Fungi on White Tea Residue Nutrients and In Vitro Rumen Fermentation Parameters"

_fermentation, doi:10.3390/fermentation8100557_

Round 1

Reviewer 1 Report

The paper compares 3 experimental settings to enhance the utilization of white tea residue as ruminant feed ingredient. The manuscript is interesting but it presents important issues in terms of noncompliance to the Journal’s rules, scientific depth, and numerous English language mistakes.

1-    The abstract is too long. It now has 330 words, while the Journal’s rules stated that:”  The abstract should be a total of about 200 words maximum.” as can be seen in this link https://www.mdpi.com/journal/fermentation/instructions

The results in the abstract shall be further summarized to the main significant findings.

Furthermore, the abstract does not properly reflect the results. It mentions that: “P.chrysosporium treatment group significantly reduced lignin content at 1 and 2 weeks” while the data in Table 2 shows an increase of lignin content after 2 weeks in all the studied conditions.

“P.” in P.chrysosporium and P.ostreatus shall not be abbreviated when they first appear in the Abstract and in the whole text of the manuscript.

2-    The paper reads more like a technical report than a scientific paper. Fundamental discussions are missing. The following points shall be thoroughly addressed:

-          In SEM data, why larger pores were formed with P.chrysosporium as compared to P.ostreatus and the dual culture? Which part of the cell wall was preferably removed to form those pores? Why the pores were small in dual culture? Could P.ostreatus have inhibited P.chrysosporium? What were the possible interactions between the 2 fungi when used together?

-          Why does lignin increase after 2 weeks while the fundamental uses of the fungi are for delignification?

-          There are already several studied comparing solid state fermentation using P.chrysosporium and P.ostreatus. How do the results in this work compare to the previously reported studies? What enzymes in each fungus played what role?

3-    Other issues include:

-          In Table 2, why the starting cell wall composition for P.chrysosporium at 0 week is so low (9.05) as compared to P.ostreatus and the dual culture (29.75)?

-          In Table 3, what do the numbers after GP i.e. 2, 4, 8 etc. stand for? They shall be explained as footnotes or as part of the table so that the data can be understandable at a glance.

-          The manuscript shall be meticulously checked for English mistakes, typos and grammatical misuses. Examples of mistakes include:

o   P. 2 l.66, “its ecofriendly” shall be “it is ecofriendly”

o   P2 l. 83, “silid-state” shall be “solid-state”

o   P.5 l.66, “three was no significant difference” should be “there was no significant difference”

o   Caption of Table 1: “ProMixmate” shall be “Proximate”

o   P.8 l.276“Butyrate acid” shall be either “butyrate” or “butyric acid”.

o   And many other mistakes.

Author Response

Dear reviewer,

We are thankful for your critical comments and thoughtful suggestions. These comments and thoughtful suggestions not only helped us with revising the manuscript but suggested some ideas for future studies. We have carefully modified the original manuscript based on these comments and suggestions. All changes were highlighted in yellow.

The English language was generally improved.

Our point-by-point responses to the reviewers’ comments/questions are as follows;

Responses to Reviewer:

Comment 1. The abstract is too long. It now has 330 words, while the Journal’s rules stated that:”  The abstract should be a total of about 200 words maximum.” as can be seen in this link https://www.mdpi.com/journal/fermentation/instructions. The results in the abstract shall be further summarized to the main significant findings.

Response: The abstract has been rewritten to address this concern. Thanks for your review

Comment 2. Furthermore, the abstract does not properly reflect the results. It mentions that: “P.chrysosporium treatment group significantly reduced lignin content at 1 and 2 weeks” while the data in Table 2 shows an increase of lignin content after 2 weeks in all the studied conditions.

Response: The abstract was modified to reflect the result correctly. Thank you.

Comment 3. “P.” in P.chrysosporium and P.ostreatus shall not be abbreviated when they first appear in the Abstract and in the whole text of the manuscript.

Response: The changes have been made. Thank you.

Comment 4. In SEM data, why larger pores were formed with P.chrysosporium as compared to P.ostreatus and the dual culture?

Response: P.chrysosporium is a typical non-selective fungus and grows very fast. Thus it consumes cell wall components simultaneously (cellulose, hemicellulose and lignin) at a very fast rate to support its metabolism compared to the others. This causes P.chrysosporium to leave behind or produce bigger pores than P.ostreatus and dual Dual. This had been indicated in the revised version of the manuscript

Comment 5. Could P.ostreatus have inhibited P.chrysosporium?

Response: On the contrary, we suspect that in the dual culture, P. chrysosporium might have inhibited the growth of P. ostreatus due to its faster colonization rate. This might have created some type of competition between the two species for which P.ostreatus was worse off. This had been indicated in the revised version of the manuscript.

Comment 6. What were the possible interactions between the 2 fungi when used together?

Response: Interaction studies regarding the 2 fungi weren’t considered in this study. However, possible interactions that could explain certain observations and results were incorporated in the revised manuscript.

Comment 7. Which part of the cell wall was preferably removed to form those pores?

Response: Based on the SEM, we cannot accurately determine which part of the cell wall was preferably removed to form those pores, as the SEM imagery in this particular study only gave us an idea of the superficial change in the cell wall that occurred due to fungi treatment. We could rather speculate to some extent based on available literature.

Comment 8. Why the pores were small in dual culture?

Response: We hold the view that this was due to a kind of interaction, thus competition which was established between fungi when co or dual-cultured. Thank you.

Comment 9. How do the results in this work compare to the previously reported studies?

Response: Results in this work compared to the previously reported studies have been elaborated in the revised. Thank you.

Comment 10. What enzymes in each fungus played what role?

Response: Enzymes study was not considered an objective in this particular study. However, some levels of enzyme literature that relate to and help enrich this study were introduced in the modified version of this manuscript. Thank you.

Comment 11. In table 2, why the starting cell wall composition for P. chrysosporium at 0 week is so low (9.05) as compared to P. ostreatus and dual culture (29.75)?

Response: P. chrysosporium at 0 week (9.05) is the ADL data. P. ostreatus and dual culture (29.75) are the data for ADF. Thank you.

Comment 12. In Table 3, what do the numbers after GP i.e. 2, 4, 8 etc. stand for? They shall be explained as footnotes or as part of the table so that the data can be understandable at a glance.

Response: Table 3 content was modified to address the concern raised. Thank you.

Comment 13. P. 2 l.66, “its ecofriendly” shall be “it is ecofriendly”

Response: The correction has been done. Thank you.

Comment 14. P2 l. 83, “silid-state” shall be “solid-state”

Response: The correction has been done. Thank you.

Comment 15. P.5 l.66, “three was no significant difference” should be “there was no significant difference”.

Response: The modification has been done. Thank you.

Comment 16. Caption of Table 1: “ProMixmate” shall be “Proximate”

Response: The error was rectified. Thank you.

Comment 17.P.8 l.276“Butyrate acid” shall be either “butyrate” or “butyric acid”.

Response: The changes was made. Thank you.

Reviewer 2 Report

Effects of solid-state fermentation pretreatment with single or dual culture white rot fungi on white tea residue nutrients and in vitro rumen fermentation parameters

The manuscript is written well in overall judgment. Some minor corrections are required before accepting this article that I stated below. This work has scientific merit to be published din Fermentation.

Abstract

L32: 1 week solid-state fermentation 32 of white tea residue using P.chrysosporium was the most desirable . >> One week solid-state fermentation 32 of white tea residue using P.chrysosporium was the most desirable.

Introduction

L52: instead of (http://www.fao.org/faostat/en/data/QCL) better to cite (FAO) and then reference it at the refence section

L134: Please provide the citation for Van Seost

M&M

Please provide specifications of the cows used (e.g. BW, parity, etc.)

Results

Stated well and in details

Discussion

L299: Nazri nayana (27) is wrongly spelled and cited. Please correct both in the text and in the reference section

L301-302: Another reason could be, may be…

Please re-phrase. Authors should speculate or attribute their result using other references

L343: After Niu et al. [36] full stop is needed.

L359: What “it” refer to? Please specify

Author Response

Dear reviewer,

We are thankful for your critical comments and thoughtful suggestions. These comments and thoughtful suggestions not only helped us with revising the manuscript but suggested some ideas for future studies. We have carefully modified the original manuscript based on these comments and suggestions. All changes were highlighted in yellow.

The English language was generally improved.

Our point-by-point responses to the reviewers’ comments/questions are as follows;

Responses to Reviewer:

Comment 1. L32: 1 week solid-state fermentation 32 of white tea residue using P.chrysosporium was the most desirable . >> One week solid-state fermentation 32 of white tea residue using P.chrysosporium was the most desirable.

Response: The sentence has been rewritten. Thank you.

Comment 2. L52: instead of (http://www.fao.org/faostat/en/#data/QCL) better to cite (FAO) and then reference it at the reference section.

Response: We have made changes and added the website to the references. Thank you.

Comment 3. Please provide the citation for Van Seost.

Response: The citation by Van Seost has been added.

Comment 4. Please provide specifications of the cows used (e.g. BW, parity, etc.)

Response: The information have been provided. Thank you.

Comment 5. L299: Nazri nayana (27) is wrongly spelled and cited. Please correct both in the text and in the reference section.

Response: The correction has been made. Thank you.

Comment 6. L301-302: Another reason could be, may be… Please re-phrase. Authors should speculate or attribute their result using other references.

Response: We have made the changes to the sentence as suggested. Thank you.

Comment 7. L343: After Niu et al. [36] full stop is needed.

Response 8: The error has been rectified. Thank you.

Comment 9. L359: What “it” refer to? Please specify

Response: “it” refers to fungi and the modification has been made in the article. Thank you.
